# NON-INHERENT FEATURE COMPATIBLE LEARNING

## ABSTRACT

The need of Feature Compatible Learning (FCL) arises from many large scale retrieval-based applications, where updating the entire library of embedding vectors is expensive. When an upgraded embedding model shows potential, it is desired to transform the benefit of the new model without refreshing the library. While progresses have been made along this new direction, existing approaches for feature compatible learning mostly rely on old training data and classifiers, which are not available in many industry settings. In this work, we introduce an approach for feature compatible learning without inheriting old classifier and training data, *i.e.*, Non-Inherent Feature Compatible Learning. Our approach requires only features extracted by *old* model's backbone and *new* training data, and makes no assumption about the overlap between old and new training data. We propose a unified framework for FCL, and extend it to handle the case where the old model is a black-box. Specifically, we learn a simple pseudo classifier in lieu of the old model, and further enhance it with a random walk algorithm. As a result, the embedding features produced by the new model can be matched with those from the old model without sacrificing performance. Experiments on ImageNet ILSVRC 2012 and Places365 data proved the efficacy of the proposed approach.

## 1 INTRODUCTION

In recent years, deep learning based methods achieved huge success in various of computer vision tasks, especially for visual searching since they could provide powerful feature representations. In a typical visual search system, the deployed deep learning model extracts the features of both *gallery* and *query* images as discriminate representations. During the retrieval stage, *gallery* images will be ranked based on their feature distances (*e.g.* Euclidean distance) to *query* images. In conventional approaches, the query and gallery features are generated by the same model. Once the deployed model of retrieval system is updated, the entire set of *gallery* features need to be 'backfilled' or 're-indexed' (Shen et al., 2020). As time goes by, the *gallery* becomes extremely large and 'backfilling' could be a painful process since millions even billions of images need to be re-processed by the new model, which is computationally expensive. There has to be a new mechanism that processes *gallery* images and the query image with two different models, while still maintaining the retrieval accuracy. In other words, the new deployed model extracted features should be 'compatible' to the existing ones without sacrificing accuracy. Such feature compatible learning problem is also named as 'Backward-Compatible Training' (Shen et al., 2020), or 'Asymmetric Metric Learning' (Budnik & Avrithis, 2020).

Existing approaches for feature compatible learning assumed significant overlap between new and old training sets. In Shen et al. (2020), the training set for new embedding model is a superset of the old set. In Budnik & Avrithis (2020), the training set for large and small models is the same, which means obtaining new model in an incremental way is not possible. Besides, in Shen et al. (2020), the classifier for old model is also needed for computing the influence loss, which is a strong requirement in real applications. As an example, a model deployed in a recommendation system as a black-box API takes images as input and returns the processed features, but the parameters of the model are not accessible. In addition, its classifier and training details are not available, neither does the formula of the loss function. This kind of setting is quite common for various practical reasons in search, recommendation, content understanding and review applications.

To address the limitation, we propose an approach for non-inherent feature compatible learning, which only exploits the old model backbone and new training data. Despite the lack of old training data or old classifier, the new model extracts compatible features without sacrificing accuracy. The proposed approach has three contributions including:

- Study and formulate the non-inherent setting of the FCL problem for the first time

- Establish a baseline with a data-incremental approach, where performance degradation is prevented by regularizing the training process of the new model

- Extend the baseline with a random walk algorithm that further improves accuracy

The experiments conducted on several standard data sets validated the effectiveness of the proposed approach.

## 2   RELATED WORK

Our approach is most relevant to feature compatible learning (Shen et al., 2020; Budnik & Avrithis, 2020; Wang et al., 2020), which has drawn attention from the research community due to the increasing size of *gallery* sets, and the heavy workload of re-generating gallery features.

### 2.1   FEATURE COMPATIBLE LEARNING

Shen et al. (2020) first formulated the 'backward-compatible' problem by deriving influence loss from an empirical criterion, and solved it by utilizing the old model to regularize the optimization process.

In Budnik & Avrithis (2020), authors investigated the problem of asymmetric test, where the *gallery* images are represented by a teacher model and *query* images are represented by a student model. A pair-based metric for instance-level image retrieval was proposed to achieve the goal. In Wang et al. (2020), authors proposed Residual Bottleneck Transformation (RBT) blocks for feature embedding transferring. Some previous works (Li et al., 2015; Yu et al., 2018) discussed the connection between features that learned by different models. However, all methods mentioned above either can not achieve the model compatibility in a data-incremental way or need to exploit old classifier for training. Our proposed method could achieve compatibility without old training data or utilizing any old classifier.

### 2.2   INCREMENTAL LEARNING

Incremental learning and Life-long learning (Rebuffi et al., 2017; Li & Hoiem, 2017) approaches aimed to stabilize model predictions when updating with new training data. Different from feature compatible problem, incremental learning focuses on maintaining performance on 'old' classes after introducing 'new' ones. In Li & Hoiem (2017), authors utilized knowledge distillation (Hinton et al., 2015) and teacher-student models for regularizing features on new data, where model distillation was used as a form of regularization when introducing new classes. In Rebuffi et al. (2017), authors proposed to use old class centers to regularize the model learning when new classes were introduced. Most approaches of incremental learning focus on the stability of classifier output, while feature compatible learning try to solve the feature compatibility problem among different models.

### 2.3   EMBEDDING LEARNING

Our approach is also relevant to the embedding learning problem that optimizes a distance metric to improve discriminative power and robustness of embedded features. There were efforts on designing powerful network architectures (He et al., 2016; Simonyan & Zisserman, 2014; Szegedy et al., 2015), discriminative loss functions (Deng et al., 2019; Wang et al., 2018), and model optimizers (Kingma & Ba, 2015), but few of them set FCL as a part of their objectives.

## 3 APPROACH

This section re-visits the basic criterion for feature compatible learning, and presents the formulation and solutions to the Non-Inherent Feature Compatible Learning problem.

### 3.1 FORMULATION AND CRITERION FOR FEATURE COMPATIBLE LEARNING

Following the FCL formulation in learning (Shen et al., 2020; Budnik & Avrithis, 2020), an old embedding model $\phi_{\text{old}}$ trained on old training set $\mathcal{D}_{\text{old}}$ maps an image $x$ to a feature vector $f = \phi_{\text{old}}(x)$, $f \in R^{K_{\text{old}}}$, where $K_{\text{old}}$ is the feature vector dimension of $f$. After a period of time, a new embedding model $\phi_{\text{new}}$ trained on $\mathcal{D}_{\text{new}}$ is obtained. The new embedding model $\phi_{\text{new}}$ maps the image $x$ into a feature vector $f$ with dimension of $K_{\text{new}}$, where $K_{\text{old}}$ and $K_{\text{new}}$ are not necessarily equal. In our setting, as explained in Sec.1 , the old training set $D_{\text{old}}$ is disjoint with the new training set $D_{\text{new}}$ ($D_{\text{old}} \cap D_{\text{new}} = \emptyset$). In addition the old classifier, training details, and the loss function are not available.

The *gallery* set is defined as $\mathcal{G}$. Since the amount of images in $\mathcal{G}$ could be extremely large, the images in $\mathcal{G}$ are usually presented as extracted feature vectors by an old embedding model, $\phi_{\text{old}}$, to save the test time and a query image will be presented as a feature vector with the same embedding model. However, once the embedding model is updated, all the images in $\mathcal{G}$ must be presented by the new embedding model $\phi_{\text{new}}$, leading to an expensive refresh of the index. To avoid such computation, we need to make it feasible to directly compare $\mathcal{G}$'s embeddings produced by $\phi_{\text{old}}$ with the query embedding by $\phi_{\text{new}}$.

In Shen et al. (2020), a strict criterion for feature backward compatible is defined as,

$$
\begin{aligned}
d(\phi_{\text{new}}(x_i), \phi_{\text{old}}(x_j)) &\geq d(\phi_{\text{old}}(x_i), \phi_{\text{old}}(x_j)), \\
&\quad \forall (i,j) \in \{(i,j)|y_i \neq y_j\}. \\
\text{and,} & \\
d(\phi_{\text{new}}(x_i), \phi_{\text{old}}(x_j)) &\leq d(\phi_{\text{old}}(x_i), \phi_{\text{old}}(x_j)), \\
&\quad \forall (i,j) \in \{(i,j)|y_i = y_j\},
\end{aligned}
\tag{1}
$$

where the $d(\cdot, \cdot)$ is the Euclidean distance metric, $x_i$ and $x_j$ are the $i^{\text{th}}$ query image and $j^{\text{th}}$ gallery image respectively, $y_i$ and $y_j$ are their labels. Such a strict criterion needs every query-gallery pair fulfills the distance requirement, which is not feasible. Alternatively, authors in Shen et al. (2020) proposed a empirical criterion as the following:

$$
M(\phi_{\text{new}}, \phi_{\text{old}}) > M(\phi_{\text{old}}, \phi_{\text{old}}).
\tag{2}
$$

where $M$ is a evaluation metric for the corresponding retrieval test set, consisting of gallery set $\mathcal{G}$ and query set. $M(\phi_{\text{new}}, \phi_{\text{old}})$ means to extract query features with new model $\phi_{\text{new}}$ and gallery features with old model $\phi_{\text{old}}$, which is defined as **cross test**. Such a criterion requires the cross test performance between the new and the old model must be better than the test using only the old model, which is defined as **self test**.

To achieve the backward compatibility defined by Eq. 2, authors in Shen et al. (2020) proposed as below an influence loss term to add into the loss function for regularizing the new embedding model training:

$$
L_{\text{BCT}}(w_c, w_\phi; \mathcal{D}_{\text{new}}) = L(w_c, w_\phi; \mathcal{D}_{\text{new}}) + \lambda L(w_{c\text{ old}}, w_\phi; \mathcal{D}_{\text{old}}),
\tag{3}
$$

where $L$ is the loss function (normally a softmax-like loss), $w_c$, $w_{c\text{ old}}$, and $w_\phi$ denote the classifier weights of the new embedding model, the classifier weights of the old embedding model, and the weights of the new embedding model's backbone, respectively. The influence loss is the second term in Eq. 3, which fixes $w_{c\text{ old}}$ and regularizes the feature output of $\phi_{\text{new}}$ to be compatible with $\phi_{\text{old}}$, generating relatively small loss on the old classifier. Meanwhile, the new training set $\mathcal{D}_{\text{new}}$ is a superset of $\mathcal{D}_{\text{old}}$.

Different from the setting in Shen et al. (2020), we do not assume there is overlap between the old and new embedding training set $\mathcal{D}_{\text{old}}$ and $\mathcal{D}_{\text{new}}$. Moreover, the parameters of old classifier $w_{c\text{ old}}$ are not available, which means we need to make the feature compatibility happen in a incremental way, without knowing the old classifier.

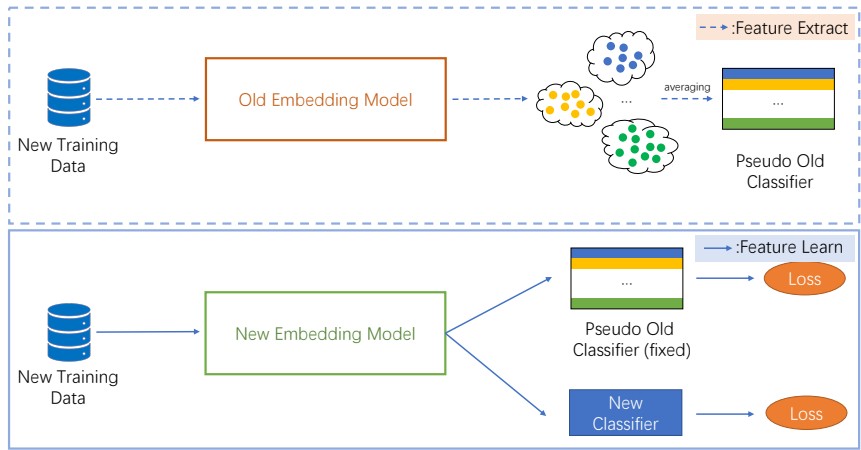

Figure 1: Illustration of using averaged features extracted by old model to serve as pseudo classifier. During the training, this pseudo classifier will be not updated by the gradient.

### 3.2 ACHIEVING NON-INHERENT FEATURE COMPATIBLE LEARNING

In this section, we will introduce our proposed feature backward training scheme without using old model's classifier $w_{c \, \text{old}}$ or old training data $\mathcal{D}_{\text{old}}$. Although we could not obtain the 'old' classifier on old training data, which is essential for influence loss in Eq. 3, we can still try to generate a 'pseudo' old classifier. Inspired by Rebuffi et al. (2017), Wu et al. (2018) and Xiao et al. (2017), we found that the classifier weights actually act as feature embedding centers so that they can generate large score with positive images' features, which suggests that even simply averaging images features processed by $\phi_{\text{old}}$ in each class, we could still obtain a reasonable classifier weight for that class, which could be denoted as,

$$w_{c \, \text{old}}^{\text{a-pse}}(n) = \frac{1}{m} \sum_{i \in \mathcal{D}_{\text{new}}(n)} \phi_{\text{old}}(x_{ni}),$$ (4)

where $m$ is the image amount in $n-$th class, $i$ is the $i$-th image in that class, $\phi_{\text{old}}$ is the old embedding model. $w_{c \, \text{old}}^{\text{a-pse}}(n)$ could be served as $n-$th column in the pseudo classifier. In practice, we found that using averaged feature vectors as classifier may not converge, classifier consisted of normed feature vector could lead to a much better result,

$$w_{c \, \text{old}}^{\text{an-pse}}(n) = \frac{w_{c \, \text{old}}^{\text{a-pse}}(n)}{\|w_{c \, \text{old}}^{\text{a-pse}}(n)\|}.$$ (5)

Another advantage of the proposed approach is that the old training set is not compulsory to use. The 'pseudo' classifier for $\mathcal{D}_{\text{new}}$ could be generated with the old embedding model easily, no data in $\mathcal{D}_{\text{old}}$ is required, then the loss function in Eq. 3 becomes to

$$L_{\text{BCT}}(w_c, w_\phi; \mathcal{D}_{\text{new}}) = L(w_c, w_\phi; \mathcal{D}_{\text{new}}) + \lambda L(w_{c \, \text{old}}^{\text{an-pse}}, w_\phi; \mathcal{D}_{\text{new}}).$$ (6)

### 3.3 RANDOM WALK REFINEMENT FOR PSEUDO CLASSIFIER GENERATION

#### 3.3.1 LIMITATION OF BASELINE APPROACH

The baseline method proposed in Section 3.2 exploits the old embedding model and new training set for 'pseudo' classifier generation and it can provide us a reasonable result for non-inherent feature compatible learning without 'old' classifier. However, due to mis-labeling or low quality, there are some outlier images in each class, simply averaging the feature vectors extracted by old embedding could not eliminate their negative effects. In this section, we will present our random walk approach for refined pseudo classifier generation.

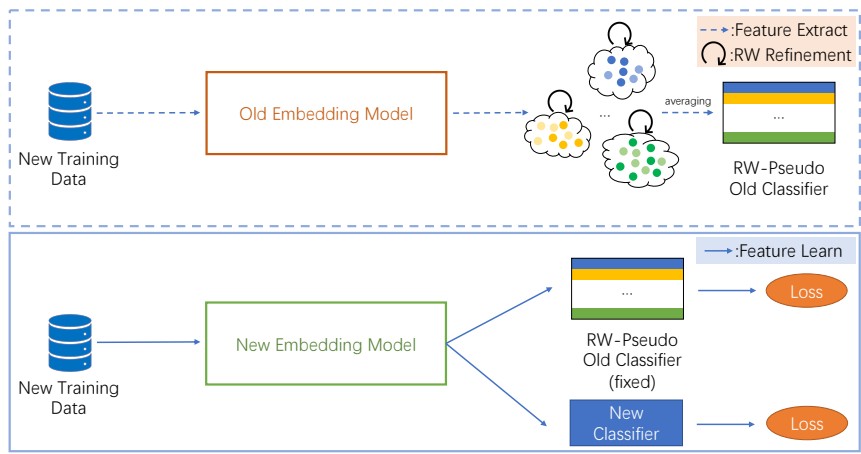

Figure 2: Illustration of using random walk algorithm to refine features extracted by old model before averaged to serve as a column of pseudo classifier. Negative effects of outliers in each class will be reduced. During the training, this pseudo classifier will be not updated by the gradient.

### 3.3.2 RANDOM WALK FOR CLASSIFIER GENERATION REFINEMENT

Random walk algorithm (Aldous, 1989) has been widely used for ranking system (Page et al., 1999) or retrieval results refinement (Loy et al., 2013; Bertasius et al., 2017). Random walk is operated on a fully connected undirected graph $G = (V, E)$, where $V$ is the vertice and $E$ is the edge. The similarities between different vertices could be defined as a symmetric matrix $S$. Each element on the matrix, $S(i, j)$, represents the similarity between vertices $i$ and $j$ (e.g. normalized cosine similarity).

In our context, in $n$-th class, $i$-th image's feature $f_{ni} \in \mathbb{R}^K$ could be represented as a vertice $V$. All of the images' features in one class can be concatenated as a matrix $F_n \in \mathbb{R}^{K \times m}$, where $m$ is the image amount of $n$-th class. The edges, $E$, among vertices are the similarity scores between different vertices. With notations mentioned above, random walk operation could be denoted as,

$$F_n^t = F_n^{t-1} S, \tag{7}$$

where $t$ is the iteration times, $S \in \mathbb{R}^{m \times m}$ is the normalized similarity matrix between different images, which is denoted as,

$$S'(i, j) = \begin{cases} \frac{\exp(S(i,j)/T)}{\sum_{j \neq i} \exp(S(i,j)/T)}, & i \neq j \\ 0, & i = j \end{cases}, \tag{8}$$

where lower temperature $T$ leads to a more concentrate probability distribution.

In practice, the refinement by random walk also needs to be weighted with the initial feature matrix $F_n^0$, which is computed as,

$$F_n^t = \lambda F_n^{t-1} S' + (1 - \lambda) F_n^0, \tag{9}$$

where $\lambda \in [0, 1]$ is the weight parameter.

If $t$ tends to infinity, the Eq. 9 has a converaged close form, which is,

$$F_n^\infty = (1 - \lambda) F_n^0 (I - \lambda S')^{-1}, \tag{10}$$

where $I$ is the identity matrix and $(\ )^{-1}$ denotes matrix inverse operation.

Once the $F_n^\infty$ is obtained, the $w_{c\ \text{old}}^{\text{rw}-\text{pse}}(n)$ could be computed with column-wised average pooling

$$w_{c\ \text{old}}^{\text{rw}-\text{pse}}(n) = \frac{1}{m}\sum_{i=1}^{m} F_n^{\infty}(:,i). \tag{11}$$

The refined classifier is also column-wised normalized like Eq. (5), which is denoted as $w_{c\ \text{old}}^{\text{rwn}-\text{pse}}$ and the influence loss changes to,

$$L_{\text{BCT}}(w_c, w_\phi; \mathcal{D}_{\text{new}}) = L(w_c, w_\phi; \mathcal{D}_{\text{new}}) + \lambda L(w_{c\ \text{old}}^{\text{rwn}-\text{pse}}, w_\phi; \mathcal{D}_{\text{new}}). \tag{12}$$

Compared with baseline method introduced in Section 3.1, our proposed random walk refinement approach incorperates more information among images' feature within one class to conduct the pseudo classifier generation, which could largely eliminate the outlier effects. In the experiments, we further show the effectiveness of the proposed method.

## 4 EXPERIMENTS

We validate the effectiveness of the proposed Non-Inherent Feature Compatible Learning approach on two large public datasets. In this section, We firstly demonstrate the datasets we use and then illustrate the baseline method performance and the effectiveness of the proposed random walk based method. Finally, we analysis the influence of hyperparameter choosing.

### 4.1 DATASETS AND METRIC

We utilize ImageNet ILSVRC (Deng et al., 2009) and Place365 (Zhou et al., 2017) datasets for training and evaluation. ImageNet dataset contains more than 1.2 million of images with 1,000 classes, Place365 contains about 1.8 million images with 365 classes. We conduct the retrieval process on these two datasets' validation sets, which means each image will be considered as a query image and all other images will be considered as gallery images. The Euclidean distance is used for ranking. The evaluation metric is Top-1 and Top-5 accuracies. We use the final global averaged pooled feature (before feed into classifier) for the feature distance computing. When conducting the cross test, both old and new model will be utilized for extracting features and distance metric will be computed with old and new feature. If dimensions of old and new features are different, zero padding will be used for dimension alignment. Following the backward-compatibility measuring setting in Shen et al. (2020), we consider backward compatibility achieved when the cross test accuracy using new model for queries and old model for galleries surpasses the self test result of only using old model.

### 4.2 TRAINING DETAILS

We use 4 NVIDIA P40 GPUs for training. For each class on the training set, 30% of training data will be used for old model training and 70% data will be used for new model training. We adopt two widely used backbones, ResNet-18 and ResNet-50 (He et al., 2016), to serve as old model and new model, the output feature dimensions of them are 512 and 2048 respectively. The image input size is set to $224 \times 224$. Only random crop and random flip are used for data augmentation. We adopt standard stochastic gradient descent (SGD) to optimize the model parameters. The learning rate is set to 0.1 and decreases 10 times every 30 epochs and the training stops after 90 epochs. The weight decay is set to $10^{-4}$ and momentum is 0.9. The weights of two terms in Eq. 3 are equal. The batch size is set to 1024. Both old and new model are random initialized (from scratch). For random walk based refinement, we set $\lambda$ to 0.9.

### 4.3 BASELINE AND PROPOSED APPROACHES ANALYSIS

In this section, we conduct several naive baseline approaches for feature compatible learning without old classifier and old training data. The cross test between old model and these approaches trained new models verified the effectiveness of the method. Besides, surprisingly, the self test of these proposed method even outperform the new model trained without any regularization.

| Old Model | New Model | Cross Test | | | | Self Test | | | |
|---|---|---|---|---|---|---|---|---|---|
| | | ImageNet | | Places365 | | ImageNet | | Places365 | |
| | | top-1 | top-5 | top-1 | top-5 | top-1 | top-5 | top-1 | top-5 |
| $\phi_{\text{old 30\%}}$ | | - | - | - | - | 39.6 | 62.7 | 28.2 | 57.2 |
| $\phi_{\text{old 30\%}}$ | $\phi_{\text{new 70\%}}$ | 0.1 | 0.5 | 0.0 | 0.2 | - | - | - | - |
| $\phi_{\text{old 30\%}}$ | $\phi_{\text{new 70\%}}^{L_2}$ | 39.4 | 62.5 | 28.2 | 57.3 | - | - | - | - |
| $\phi_{\text{old 30\%}}$ | $\phi_{\text{new 70\%}}^{\text{t-pse}}$ | 30.6 | 57.6 | 30.7 | 60.3 | - | - | - | - |
| $\phi_{\text{old 30\%}}$ | $\phi_{\text{new 70\%}}^{\text{an-pse}}$(Ours) | 55.0 | 77.6 | 36.9 | 63.8 | - | - | - | - |
| $\phi_{\text{old 30\%}}$ | $\phi_{\text{new 70\%}}^{\text{rwn-pse}}$(Ours) | 56.2 | 77.8 | 37.4 | 64.0 | - | - | - | - |
| $\phi_{\text{new 70\%}}$ | | - | - | - | - | 58.4 | 78.7 | 33.8 | 62.4 |
| $\phi_{\text{new 70\%}}^{L_2}$ | | - | - | - | - | 41.1 | 64.1 | 29.0 | 58.3 |
| $\phi_{\text{new 70\%}}^{\text{t-pse}}$ | | - | - | - | - | 61.3 | 80.0 | 35.5 | 64.4 |
| $\phi_{\text{new 70\%}}^{\text{an-pse}}$ (Ours) | | - | - | - | - | 61.9 | 80.3 | 38.9 | 65.7 |
| $\phi_{\text{new 70\%}}^{\text{rwn-pse}}$ (Ours) | | - | - | - | - | 62.2 | 80.4 | 38.2 | 65.9 |

Table 1: Baseline and proposed approaches comparison on ImageNet Deng et al. (2009) and Places365 Zhou et al. (2017) datasets. The self test results of $\phi_{\text{old 30\%}}$ is the lower bound of the comparison.

**No regularization between $\phi_{\text{new}}$ and $\phi_{\text{old}}$** We firstly test a simple case which was verified in Shen et al. (2020), directly comparing the isolatedly trained new model (denoted as $\phi_{\text{new 70\%}}$) and old model (denoted as $\phi_{\text{old 30\%}}$). As illustrated in Section 4.2, we use 30% of training data in each class to train $\phi_{\text{old 30\%}}$ and 70% of data is used for training $\phi_{\text{new 70\%}}$. For aligning their output feature dimensionality, we directly use zero-padding for $\phi_{\text{old 30\%}}$'s feature output to pad it from 512 to 2048. The results on ImageNet and Places365 are shown in Table 1, such directly comparison is an epic failure, which has already been verified in Shen et al. (2020).

$L_2$ **regression between $\phi_{\text{new}}$ and $\phi_{\text{old}}$ output feature** A simple baseline that could be proposed is using $L_2$ loss to minimize the output features' Euclidean distance between 'old' and 'new' model, which is denoted as $\phi_{\text{new 70\%}}^{L_2}$ in Table 1. Such simple baseline could not meet the requirement of the feature compatible learning. The possible reason for $L_2$ loss failure is that $L_2$ loss only focuses on decreasing distance between feature pairs from the same image, however, it ignores the distance restriction between negative pairs.

**Trained pseudo classifier with new data and $\phi_{\text{old}}$ output feature** Another baseline we compared is using old model backbone to process new training data for obtaining the corresponding features, then these features will be fed into a linear layer. The output of this linear layer will be sent into a softmax layer and optimized with cross-entropy loss and new training data label. The loss function could be denoted as,

$$L = -\frac{1}{N}\sum_{i\in\mathcal{D}_{\text{new}}}\log\big(\frac{\exp(w^T f_i[L_i])}{\sum_j \exp(w^T f_i[j])}\big),\tag{13}$$

where $f_i$ is the $i$-th image's feature processed by $\phi_{\text{old 30\%}}$, $L_i$ is the corresponding label, $N$ is the new training data amount, $w$ is the linear classifier weight.

With minimizing the loss denoted in Eq. 13, we can obtain a linear classifier $w$ which produces relatively small loss with fixed old backbone $\phi_{\text{old 30\%}}$ on new training data, which could be used in influence loss (Eq. 3) to serve as $w_{\text{old}}$. The 'new' model trained with such regularization is denoted as $\phi_{\text{new 70\%}}^{\text{t-pse}}$, the cross test result in Table 1 shows that although it is not a total failure, but they can not beat the self test result of $\phi_{\text{old 30\%}}$, which is the basic requirement for feature compatible learning.

| Old Model | New Model | $T$ | $\lambda$ | ImageNet | |
|---|---|---|---|---|---|
| | | | | top-1 | top-5 |
| | | 1.0 | 0.1 | 55.7 | 77.8 |
| | | 1.0 | 0.5 | 55.9 | 77.9 |
| $\phi_{\text{old 30\%}}$ | $\phi_{\text{new 70\%}}^{\text{rw-pse}}$ | 1.0 | 0.9 | 55.7 | 78.0 |
| | | 0.1 | 0.9 | 56.1 | 77.9 |
| | | 0.05 | 0.9 | 56.2 | 78.1 |

Table 2: The hyperparameter analysis about softmax normalization temperature and random walk weight parameter.

**Feature averaging pseudo classifier**  As illustrated in Section 3.2, we could make a pseudo classifier with averaged feature representation of all images in each class on new training dataset. Then we utilize such pseudo classifier to regularize the new model training as shown in Eq. 6. The model is denoted as $\phi_{\text{new 70\%}}^{\text{an-pse}}$. As illustrated in Table 1, such simple approach could actually satisfy the feature compatible learning requirement, which surpasses the self test performance of using $\phi_{\text{old 30\%}}$ by 15.4% in terms of top-1 accuracy on ImageNet dataset and 8.7 % on Places365 dataset.

**Feature averaging pseudo classifier with random walk refinement**  Following the algorithm introduced in Section 3.3, we conduct random walk refinement for generating classifier, we set temperature parameter $T$ in Eq. 8 to 0.05 and the $\lambda$ in Eq. 7 to 0.9. Then the refined classifier $w_{c\ \text{old}}^{\text{rwn-pse}}$ will be served as old classifier in Eq. 12. The trained new model with such regularization is denoted as $\phi_{\text{new 70\%}}^{\text{rwn-pse}}$. As illustrated in Table 1, the proposed random walk refinement approach not only fulfill the requirement of feature compatible training, but also outperforms all compared methods.

**Better self test performance**  In Table 1, we show that with our proposed method, the self test results of $\phi_{\text{new 70\%}}^{\text{an-pse}}$ and $\phi_{\text{new 70\%}}^{\text{rwn-pse}}$ even surpass the performance of $\phi_{\text{new 70\%}}$, which means with the proposed approach, the feature could be improved with the information that provided by old model.

### 4.4 Hyperparameters Analysis for Random Walk

**Temperature $T$ setting for softmax normalization**  We firstly investigate the influence of softmax normalization temperature in Eq. 8. High temperature in softmax function leads to a softer distribution, while based on our analysis in Section 3.3, there are some noisy images in each class. We need to make the averaged feature more concentrate on the non-noisy images, so the temperature of the softmax should be low. As illustrated in Table 2, such assumption is verified. Lower temperature provides better results.

**Effect of choosing different $\lambda$**  Another investigation is about the choice of $\lambda$ in Eq. 10, larger $\lambda$ means the larger weight for random walk refined feature and less for original feature. As shown in Table 2, larger $\lambda$ will provide slightly better performance.

## 5 Conclusion

In this paper, we investigate several approaches to achieve non-inherent feature compatibility, which aims at achieving feature compatibility without utilizing old training data and old classifier. We conduct comparison experiments on two large public datasets. Compared with other simple baselines, our approach achieved better performance with limited offline computing cost increasing. Furthermore, the proposed method could even improve the feature learning and achieves better self test result compared with model trained without any feature compatible regularization. In the future, we would like to investigate more about the regularization scheme for filling up the accuracy gap between cross test and self test.

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
