# OpenReview forum: "Non-Inherent Feature Compatible Learning"
_ICLR.cc/2021/Conference — Reject_

### Official Review · AnonReviewer4 · 2020-10-25
**good topic and straightforward solution, but need more explanations and comparisons**

**Rating:** 5
**Confidence:** 3

**Review:**

This paper addresses an interesting problem in retrieval system - compatible features learning. Given the old feature extractor and a new dataset, the objective is to learn a new feature extractor, so that the features extracted by two (old and new) feature extractors are comparable to each other. In the proposed setting, the old dataset (including its statistics), old classifier, and the parameters of the old model are not available.

The presentation of this paper is clear, including the problem description and basic idea of the methodology. The construction of the pseudo old classifier is well motivated and reasonable. The performance improvements are significant based on the presented results.

However, I still have some concerns.
First, it is unclear that why random walk provides benefit to obtaining a better feature matrix.
Based on eq 10, each column of the new feature matrix (left side) is the linear combination of the columns of the old feature matrix. Subsequently, the feature matrix is averaged to obtain the pseudo classifier. So intuitively, the pseudo classifier is obtained by weighted averaging of the feature matrix, where the weights are learned by random walk from their similarity scores.
Given these facts, more explanations are needed for why random walk could produce the beneficial weights.

Second, the formulation of the loss function (L) is not given. (eq 3), which is an important detail.

Third, there is no experimental comparison with other works. So it is hard to see the contribution of this paper.
I understand that this is a novel problem, which may not have prior work to compare with. However, to make the results more convincing, the proposed methods can be used to other similar settings for evaluation, e.g. the setting in Shen et al. 2020.

---

> ### Author Response · Authors · 2020-11-20
> **To R4**
>
>
> **Q12:  Why random walk could produce the beneficial weights?**
>
> **A12:** With random walk based refinement, each column of the feature matrix is averaged based on the sample-pairwise similarities, which is a message-passing process on a fully connected graph and such process is conducted many times (infinite times with close form). Each column is the node on the graph and pairwise similarities are edges. During the message passing process, the outlier feature weight is reduced gradually since they may have low similarities with other samples, which can be taken as a 'soft' version of filtering outliers. We also conduct a comparison between the random walk based refinement and 'hard' discarding filtering, which is listed in **A4**, the results show that random walk based method outperforms other compared baseline methods.
>
> **Q13: the formulation of the loss function (L) in Eq (3) is not given.**
>
> **A13:** The loss function consists of two parts, In Eq(3), the first part is the classification cross entropy loss  computed by the new model's output feature and new classifier, which is,
>
> $$
>   L(w_{c \ \rm},  w_\phi;  {\cal D_{\rm new}}) = -\frac{1}{N}\sum_{i \in {\cal D_{\rm new}}}{\rm log}(\frac{{\rm exp}(w_{c \ \  \rm}^T  w_\phi(x_i)[l_i])}{\sum_{j}{\rm exp}(w_{c \ \  \rm}^T w_\phi(x_i)[l_j])}).
> $$
>
> The second part is also a classification cross entropy loss computed by the new model's output feature and fixed old classifier, which is,
>
> $$
>   L(w_{c \ \rm old},  w_\phi;  {\cal D_{\rm old}}) = -\frac{1}{N}\sum_{i \in {\cal D_{\rm old}}}{\rm log}(\frac{{\rm exp}(w_{c \ \  \rm old}^T  w_\phi(x_i)[l_i])}{\sum_{j}{\rm exp}(w_{c \ \  \rm old}^T w_\phi(x_i)[l_j])}).
> $$
>
> Please be noted that the loss function in Eq(3) is proposed in Shen et.al. 2020, which is different from our proposed loss function, for more details about our proposed loss function, please see **A8**.
>
> **Q14: No experimental comparison with other works. e.g. the setting in Shen et al. 2020.**
>
> **A14:** For the comparison with other incremental learning methods, please see **A7**. For the comparison with Shen et.al 2020, Following the setting in Shen et.al. 2020, we generate pseudo classifiers with 50% training data  (50% IDs) on ImageNet, the feature extractor is 50% data trained model. The results are shown below,
>
> | Old Model |  New Model |  Top-1 Acc | Top-5 Acc |
> | :-----------: | :-----------: | :-----------: | :-----------: |
> |$\phi_{\rm old \  \ 50 \\%} $|$\phi_{\rm old \  \ 50 \\%} $  | 39.5%| 60.0%|
> |$\phi_{\rm old \  \ 50 \\%} $|$\phi_{\rm new \  \  100 \\%}^{\rm BCT} $|42.2%|65.5% |
> |$\phi_{\rm old \  \   50 \\%}$ |$\phi_{\rm new \  \ 100 \\%}^{\rm an-pse}$| 42.4%| 64.7%|
> |$\phi_{\rm old \  \ 50 \\%}$ |$\phi_{\rm new \  \ 100 \\%}^{\rm rwn-pse}$ |44.1% |65.0% |
> |$\phi_{\rm new \  \ 100 \\%}$ |$\phi_{\rm new \  \ 100 \\%}$ | 62.5%| 81.5%|
>
> As the table's demonstration, compared with $\phi_{\rm new \  \  100 \\%}^{\rm BCT} $ proposed by Shen et.al. 2020, our proposed method  $\phi_{\rm new \  \ 100 \\%}^{\rm rwn-pse}$  improves top-1 accuracy by 1.9% compared with $\phi_{\rm new \  \  100 \\%}^{\rm BCT} $  and 1.7% compared with $\phi_{\rm new \  \ 100 \\%}^{\rm an-pse}$, which demonstrates that the random walk refined generated classifier is more suitable for feature compatible training.
>
> Besides, the problem we wish to solve in this paper is different from the problem in Shen et.al 2020. In Shen et.al 2020, the old training data is not available and the old model hyper-parameter information is 100% transparent to users. However, we aim at making feature compatible in an incremental learning way and the information of the old model backbone part is completely invisible. (For more details, please see **A2**. ). So these two approaches are not comparable.

---

### Official Review · AnonReviewer2 · 2020-10-27
**An interesting problem**

**Rating:** 5
**Confidence:** 4

**Review:**

This paper deals with an interesting problem of feature compatible learning that the features produced by new model should be compatible with old features. The proposed method uses nearest class–mean classifier instead of linear classifier. Random walk is applied to refine the class means. The proposed method is compared with several baseline methods and shows good performance.


Pro:
1.	This paper deals with an interesting problem of feature compatible learning.
2.	A new experimental setting is presented.
3.	A simple method based on nearest class–mean classifier is proposed and shows good performance.

Cons:
1.	Some incremental learning methods also deal with the feature compatible problem when old features are stored for rehearsal. The paper lacks the review of these methods.
2.	The regularization term in Eq (6) and (12) is not defined.
3.	It needs more details to explain the two important settings (cross test and self test) and the experimental protocols.
4.	As the proposed method needs to compute the similarity graph for data in every class. How much extra time and memory does it take to implement the random walk refinement?
5.	More experimental results, e.g. visualization, should be given.

---

> ### Author Response · Authors · 2020-11-20
> **To R2**
>
> **Q7: Some incremental learning methods also deal with the feature compatible problem.**
>
> **A7:** The averaged classifier generation method could be considered as an implementation of "iCaRL: Incremental Classifier and Representation Learning", which is denoted as $\phi_{\rm new \ 70\\%}^{\rm an-pse}$. We also implement another classic incremental learning baseline method, named "Learning without Forgetting" (Li et.al. 2017 PAMI), which is denoted as $\phi_{\rm new \  \  70 \\%}^{\rm lwf} $. It utilized the old model and old classifier to output the predicted probability of new data to serve as labels. These two papers are reviewed in the Related Works part. The performance of these methods on ImageNet dataset is shown below,
>
> | Old Model |  New Model |  Top-1 Acc | Top-5 Acc |
> | :-----------: | :-----------: | :-----------: | :-----------: |
> |$\phi_{\rm old \  \   30 \\%}$ | $\phi_{\rm new \  \ 70 \\%}^{\rm an-pse}$| 55.0% | 77.6% |
> |$\phi_{\rm old \  \ 30 \\%} $|$\phi_{\rm new \  \  70 \\%}^{\rm lwf} $  | 41.4% | 66.3% |
> |$\phi_{\rm old \  \ 30 \\%}$ |$\phi_{\rm new \  \ 70 \\%}^{\rm rwn-pse}$ |56.2% | 77.8% |
>
> As shown in the above table, our proposed random walk based training method outperforms iCaRL baseline $\phi_{\rm new \  \ 70 \\%}^{\rm an-pse}$ and learning without forgetting $\phi_{\rm new \  \  70 \\%}^{\rm lwf} $ easily.
>
> **Q8: The regularization term in Eq (6) and (12) is not defined.**
>
> **A8:** The 'regularization term' is also a cross entropy loss computed by new model's feature and fixed 'old classifiier', which is denoted as,
>
> $$
>   L(w_{c \ \rm old},  w_\phi;  {\cal D_{\rm new}}) = -\frac{1}{N}\sum_{i \in {\cal D_{\rm new}}}{\rm log}(\frac{{\rm exp}(w_{c \ \  \rm old}^T  w_\phi(x_i)[l_i])}{\sum_{j}{\rm exp}(w_{c \ \  \rm old}^T w_\phi(x_i)[l_j])}),
> $$
>
> where $w_{c \ \rm old}$ is the fixed old classifier, $w_\phi$ is the new model backbone for extracting feature, $x_i$ is the $i$-th input image and $l_i$ is the corresponding label. We will add this detail to the final version.
>
> **Q9: More details about important settings (cross test and self test) and the experimental protocols?**
>
> **A9:**
> *Cross test:* Using the old model to extract gallery image features and using the new model to extract query image features. Then we compute query-to-gallery distance with these features from different sources.
>
> *Self test:* As a normal setting, we extract both query and gallery images with the same model.
>
> Please be noted that we use ImageNet and Places365 datasets, which do not have an official query-gallery set split. We simply use all of them to sever as both query and gallery and compute an N by N (N is validation set size) distance matrix for evaluation.
>
>
> **Q10:  Extra time and memory of Random Walk refinement?**
>
> **A10:** We evaluate the memory and time cost for generating a pseudo classifier on the ImageNet dataset. Compared with the baseline average method, with the random walk refinement, the classifier generating time on 70% ImageNet images (~ 8M samples) increased 48s (0.048s for each class) and the memory consumption increased 40Mb (400 Kb for each class). We can observe that there is only a subtle increament in runtime and storage when applying the random walk for pseudo classifier generation. Besides, such a process could be done offline, the model's online performance will be not affected at all.
>
> **Q11: More experimental results, e.g. visualization.**
>
> **A11:** Thanks for the comments, we will add more retrieval visual results in the final version.

---

### Official Review · AnonReviewer3 · 2020-10-29

**Rating:** 6
**Confidence:** 3

**Review:**

[Summary]
This work proposes a new problem setting by adding extra constraints to the Feature Compatible Learning problem. The new constraints avoid using old training data and the old model’s parameter when learning a new model. The paper gives a baseline method and its variants for the problem by generating pseudo classifiers to regularize a new model’s learning. The experiments show that the proposed method can satisfy the empirical criterion about success.

[Strengths]
1. The constraints added to the feature compatible learning are sounded and may have practical value.
2. The writing of this paper is satisfactory. The clarity is good and easy to follow. Both the related works, motivations, and technical details are clearly introduced.
3. The experiment has a reasonable set of baselines for the problem. The ablation study clearly shows how the proposed enhancement contributes to performance gain.
4. Section 3.3.2 is a new interesting way to combine the feature representation for a group of data.

[Weaknesses]
1. Figures 1 and 2 do not help understand the method’s overview nor the problem setting.
2. The gain of the RWN classifier is not significant. There are other simple ways to remove the outliers for computing the mean representations. One example is to ignore the top 10 percent of the data based on the initial mean and variance, then re-compute the mean representation. The RWN should be compared with those simple baselines.
3. There are several minor typos. Please have proofreading.

[Questions]
1. The gain of RWN looks small in Table 2. What is the variance of Table 2?

---

> ### Author Response · Authors · 2020-11-20
> **To R3**
>
> **Q3: Figures 1 and 2 do not help understand the method’s overview nor the problem setting.**
>
> **A3:** Thanks for the comments, we will modify our teaser figure with (1) adding more details. (2) re-writing the caption in the final version.
>
> **Q4: The gain of the RWN classifier is not significant.  There are other simple ways to remove the outliers, one example is to ignore the top 10 percent of the data.**
>
> **A4:** Following the reviewer's suggestion, in each class, we removed 10% of new training data based on the features' mean and variance. Then we re-compute the class center of the remaining data and use it for Feature Compatible Training, where the trained model is denoted as $\phi_{\rm new \  \  70 \\%}^{\rm rm \  \ 10 \\%} $. The results on ImageNet are as follows:
>
> | Old Model |  New Model |  Top-1 Acc | Top-5 Acc |
> | :-----------: | :-----------: | :-----------: | :-----------: |
> |$\phi_{\rm old \  \   30 \\%}$ | $\phi_{\rm new \  \ 70 \\%}^{\rm an-pse}$| 55.0% | 77.6% |
> |$\phi_{\rm old \  \ 30 \\%} $|$\phi_{\rm new \  \  70 \\%}^{\rm rm \  \ 10 \\%} $  | 55.5% | 77.6% |
> |$\phi_{\rm old \  \ 30 \\%}$ |$\phi_{\rm new \  \ 70 \\%}^{\rm rwn-pse}$ |56.2% | 77.8% |
>
> Compared with baseline method  $\phi_{\rm new \  \ 70\\%}^{\rm an-pse}$, removing 10% of the training data does not bring large improvement. Our proposed method with random walk refinement $\phi_{\rm new \  \ 70 \\%}^{\rm rwn-pse}$ also surpasses $\phi_{\rm new \  \  70 \\%}^{\rm rm \  \ 10 \\%} $ by 0.7% in terms of top-1 accuracy.
>
> **Q5: There are several minor typos. Please have proofreading.**
>
> **A5:** Thanks for the comments, we will revise them in the final version.
>
> **Q6: The gain of RWN looks small in Table 2. What is the variance of Table 2?**
>
> **A6:** We conduct the experiments on ImageNet and Places365 datasets, our random walk based method outperforms the baseline of averaged feature by 1.2% in terms of top-1 accuracy. Compared with the ablation studies in other papers [1], such improvement on ImageNet dataset is not incremental. Please be noted that we use the feature distance for evaluation. In table 2, the variance among different hyper-parameters setting is small, which demonstrates our proposed method is robust against hyper-parameters changing.
>
> [1] Unsupervised Feature Learning via Non-Parametric Instance Discrimination, Wu et.al. CVPR 2018

---

### Official Review · AnonReviewer1 · 2020-10-30
**This paper considers an interesting problem called feature compatible learning, that is, learn a new and better feature embedding model without the need for updating existing feature embedding.**

**Rating:** 2
**Confidence:** 4

**Review:**

I spot a couple of key issues with the proposed method:

1) Misinterpretation of the previous study by Shen et al. (CVPR 2020): As presented in Eq (8) of that CVPR paper, the dataset where the influence loss is applied is NOT necessarily the old training set; It  can be the new training dataset. Also, their results show that using new training set for the influence loss is slightly better than using old training set. This makes the statement that "previous works rely on old training" invalid. So Eq (3) of this submission is not complete and somewhat misleading.

2) Shen et al. (CVPR 2020) only use the classifier but not the feature model. Although this proposed work does not use the old classifier, the feature model is instead used to estimate the classifier weights. So, it is not true to claim that "the old model is a black-box" for the proposed method; Just use a different part of the old system. Besides, in deep classification model, the feature vectors and classifier weights are closely related and both reveal similar amount of information.  In particular, if a cosine classifier is used, the two can be exchanged to each other. So, this proposed model is actually leveraging the same amount of information from the old system, as compared to Shen et al. (CVPR 2020).

Overall, the two points above would make me to conclude that this work comes with fundamental flaws/drawbacks.

---

> ### Author Response · Authors · 2020-11-20
> **To R1**
>
>
> **Q1.1: Misinterpretation of the previous study, in Shen et al., the influence loss could be used on new training set.**
>
> **A1.1:** The reviewer thinks Shen et. al.'s work doesn't have to rely on the old data. Actually, Shen et.al. proposed method must use the old training data as a part of the new training data.  In Shen et.al's experiments setting part, most experiments trained $\phi_{\rm old}$ with the randomly sampled 50% IDs subset of the IMDBFace dataset and the new model is trained on the full IMDBFace dataset. Some experiments utilized exactly the same training set for training new and old models (only changes other factors like loss function or network structure).  So the new training set always contains the old training set ($D_{\rm new} \supseteq D_{\rm old}$). In other words, in Shen et al.(CVPR 2020), achieving backward compatibility relies on the old training dataset.  However, in our setting, we assume that there is NO overlap between the old and new training data set. Our work aims at achieving feature compatibility without any use of the old data, which is a quite challenging and practical case since the old training data may be inaccessible due to the privacy issue or storage limitation. Besides, the method proposed in Shen et.al. relies on the old classifier, while our proposed method does not. These two methods aim at achieving feature compatibility in different cases so they are not comparable.
>
> **Q1.2: Eq (3) of this submission is not complete and somewhat misleading.**
>
> **A1.2:** In our paper, Eq.(3) mentioned that the new training set $D_{\rm new}$ is a superset of the old training set $D_{\rm old}$, which properly demonstrates the setting in Shen et.al. (Detailed explanation is in **A1.1**). So Eq.(3) is not misleading.
>
> **Q2: It not true to claim that 'the old model is a black box'.**
>
> **A2:**  The reviewer may think we claimed the 'old system' (old model + old classifier) is a black box, which is not true. Instead, we only claimed the old model (feature extractor) is a black box since we can not access the old model's inside information (e.g. the 'old' model's training data, network structure, parameters, training loss function). In our work, only the input images and the output feature of the old model are available and other key information about the old model is hidden (The readers may have experienced this process when calling an API, the inside of API is completely invisible, only processed features are returned), which is exactly a 'black-box' case based on the definition (*black-box*  is a device, system, or object which can be viewed in terms of its inputs and outputs, without any knowledge of its internal workings). However, In Shen et.al. 2020, the 'old' model's training data, network structure, loss function are transparent to the users, which can greatly help them to design a 'proper' new model.

---

### Decision · Program_Chairs · 2021-01-07
**Final Decision**

**Decision:**

Reject

**Comment:**

This paper deals with a problem of feature compatible learning, where the features produced by new model should be compatible with old features. As pointed out by the reviewers, there are several weaknesses with this paper: (a) the novelty is not strong enough, (b) the experimental results should be better explained and be more thorough, (c) the formulation is not well motivated.